# Virus-Induced Flowering by Apple Latent Spherical Virus Vector: Effective Use to Accelerate Breeding of Grapevine

**DOI:** 10.3390/v12010070

**Published:** 2020-01-07

**Authors:** Kiyoaki Maeda, Teppei Kikuchi, Ichiro Kasajima, Chungjiang Li, Noriko Yamagishi, Hiroyuki Yamashita, Nobuyuki Yoshikawa

**Affiliations:** 1Faculty of Agriculture, Iwate University, Morioka 020-8550, Japan; g0418033@iwate-u.ac.jp (K.M.); a0116014@iwate-u.ac.jp (T.K.); cjli_xm@aliyun.com (C.L.); 2Agri-Innovation Center, Iwate University, Morioka 020-8550, Japan; kasajima@iwate-u.ac.jp (I.K.); nyamagi@iwate-u.ac.jp (N.Y.); 3Experimental Farm, Faculty of Life and Environmental Science, University of Yamanashi, Kofu 400-8510, Japan; hyamashita@yamanashi.ac.jp

**Keywords:** grapevine, apple latent spherical virus vector, virus-induced flowering, reduced generation time, breeding of grapevine, virus elimination

## Abstract

Apple latent spherical virus (ALSV) was successfully used in promoting flowering (virus-induced flowering, VIF) in apple and pear seedlings. In this paper, we report the use of ALSV vectors for VIF in seedlings and in vitro cultures of grapevine. After adjusting experimental conditions for biolistic inoculation of virus RNA, ALSV efficiently infected not only progeny seedlings of *Vitis* spp. ‘Koshu,’ but also in vitro cultures of *V. vinifera* ‘Neo Muscat’ without inducing viral symptoms. The grapevine seedlings and in vitro cultures inoculated with an ALSV vector expressing the ‘florigen’ gene (Arabidopsis *Flowering locus T*, *AtFT*) started to set floral buds 20–30 days after inoculation. This VIF technology was successfully used to promote flowering and produce grapes with viable seeds in in vitro cultures of F_1_ hybrids from crosses between *V. ficifolia* and *V. vinifera* and made it possible to analyze the quality of fruits within a year after germination. High-temperature (37 °C) treatment of ALSV-infected grapevine disabled virus movement to newly growing tissue to obtain ALSV-free shoots. Thus, the VIF using ALSV vectors can be used to shorten the generation time of grapevine seedlings and accelerate breeding of grapevines with desired traits.

## 1. Introduction

Most plant viruses consist of small genomic RNA and capsid proteins that autonomously propagate upon infection of host plant cells and then spread throughout the plant by cell-to-cell and long-distance movement [1]. Virus vector technologies use the ability of viruses to suppress plant genes to engineer gene silencing or expression of exogenous genes such as green fluorescent protein (GFP) through relatively short experimental procedures [2]. Suppression of plant gene expression with virus vectors is called virus-induced gene silencing (VIGS) [3,4,5], which is becoming a popular technique in model plants such as *Arabidopsis thaliana* and *Nicotiana benthamiana*. Unlike genetic transformation, virus vectors can be applied to a wide range of plant species [5]. In recent years, virus vectors used in major crops were reported for gene silencing or expression of exogenous genes; tobacco rattle virus vector in solanaceous crops, pea early browning virus and bean pod mottle virus vectors in legumes, barley stripe mosaic virus vector in barley and wheat, brome mosaic virus vector in rice, maize, and barley, and apple latent spherical virus and tobacco ringspot virus vectors in cucurbits, etc. [6,7,8,9,10,11,12,13,14,15,16].

Compared with virus vectors for herbaceous crops, virus vectors available to woody crops like fruit trees are limited because an efficient method for inoculation in woody plants has not been fully developed. Virus vectors stably maintained in host plants would also be needed for efficient gene silencing and gene expression in woody crops because of the long growth period of fruit trees. Attempts have also been made to develop viral vectors for citrus and grapevine trees, which are globally important crops, and several vectors have been reported, including the citrus tristeza virus (CTV) and citrus leaf blotch virus vectors to promote gene expression and gene silencing in citrus plants, respectively [17,18,19]. In grapes, silencing the phytoene desaturase (PDS) gene and GFP expression have been reported using vectors based on grapevine virus A and grapevine leafroll-associated virus-2 [20,21]. The future possibilities of viral vectors in woody crops are summarized in a review by Dawson and Folimonova [22], although the CTV vector in citrus is the main focus.

We have used the apple latent spherical virus (ALSV) vector for driving gene silencing and gene expression in various plant species including fruit trees [6,23,24,25,26]. ALSV is a spherical virus consisting of a bipartite single-stranded RNA genome (RNA1 and RNA2) and three capsid proteins (Vp25, Vp20, and Vp24) that is in the *Cheravirus* genus and the *Secoviridae* family [27]. RNA1 encodes a polyprotein for a protease cofactor, an NTP-binding helicase, a viral protein genome-linked, a cysteine protease, and an RNA polymerase. RNA2 also encodes a polyprotein for a movement protein (MP) and three capsid proteins [28,29]. The ALSV vector has several advantages as a genetic tool; for example, ALSV has a relatively broad spectrum of host plants (*Caryophyllaceae*, *Chenopodiaceae*, *Cryptomeria*, *Fabaceae*, *Cucurbitaceae*, *Gentianaceae*, *Pinus*, *Rosaceae*, *Rutaceae*, *Solanaceae*, and Arabidopsis) [6,23,24,30,31,32,33]. ALSV does not cause viral symptoms in most hosts, making it possible to evaluate the functions of silenced genes. ALSV also invades to shoot meristems and induces uniform gene silencing throughout the plant [6,24,25,34,35]. We also succeeded in promoting flowering (virus-induced flowering, VIF) in apple and pear seedlings by simultaneous expression of the Arabidopsis *Flowering locus T* gene (*AtFT*) and suppression of apple *Terminal flower 1* gene (*MdoTFL1*) [25,26,36]. Apple seedlings generally take at least 6–7 years from germination to flowering in the wild. However, using VIF, apple seedlings blossom 1–2 months after virus inoculation and the generation time was shortened to within one year. ALSV is not transferred to most next-generation seedlings [37] and the virus can be removed from infected apple and pear trees with a simple heat treatment [36].

Grapevine is one of the most popular fruit crops worldwide and global grape production currently amounts to more than 75 million tons per year [38]. In grapevines, virus-based vectors could be important biotechnological tools to improve plant disease protection and support traditional varieties used in wine making [22]. If the ALSV vector could be applied to grapevine, the technology would be very useful for basic research including gene function analysis by VIGS and in breeding of new grape varieties using VIF. 

In this research, we develop experimental procedures to efficiently infect grapevine seedlings and in vitro cultures with ALSV vectors. These vectors were successfully used for VIGS and VIF in grapevine plants. The VIF using ALSV vectors shortened the generation time of grapevine seedlings to within a year, indicating its use as a technique for accelerating grapevine breeding in combination with marker-assisted selection.

## 2. Materials and Methods

### 2.1. Plants

Progeny seeds of grapevine, *Vitis* spp. cultivar ‘Koshu’ and *V. ficifolia* var. *ganebu* (Ganebu) [39,40,41] generated by either self-pollination and F_1_ seeds from Ganebu × *V. vinifera* cv. ‘Nehelescol’ were preserved at 4 °C until use. Seeds were germinated at 25 °C on wet paper in petri dishes under illumination with fluorescent light. For preparation of in vitro cultures, F_1_ seeds of Ganebu × ‘Nehelescol’, *V. vinifera* ‘Cabernet Sauvignon’ × Ganebu, and Ganebu × *V. vinifera* ‘Shine Muscat’ crosses were germinated aseptically and grown in plant-boxes on solid medium (half-strength Murashige-Skoog salts and vitamins, 2% glucose, 0.8% agar, pH 5.8; 40 mL per each plant-box). For propagation of cultured plants, extended stems were cut every two to three nodes. In vitro cultures of cultivar ‘Neo Muscat’ kindly supplied by Dr. Ikuko Nakajima (Institute of Fruit Tree and Tea Science, NARO) were maintained on solid media as above and used for virus inoculation.

### 2.2. Construction of ALSV Vectors

The pCALSR1SM and pCLASR2XSB plasmids were used for construction of RNA1 and RNA2 ALSV vectors (Figure 1). In these vectors, RNA1 and RNA2 sequences were driven by cauliflower mosaic virus 35S RNA promoter and ended with the nopaline synthase terminator. These vectors were based on pCAMBIA1300 for introduction into *Agrobacterium tumefaciens* [36,39]. Vector structures are summarized in Figure 1. A 201-base fragment (nucleotide positions 104–304) of the *VvPDS* gene (accession No. EU816356) was synthesized and used as a template to amplify a fragment with ‘VvPDS-XhoI (+)’ and ‘VvPDS-BamHI (−)’ primers (Appendix A) to attach *Xho*I and *Bam*HI sites. Amplified DNA was digested with *Xho*I and *Bam*HI and introduced into the X/S/B site of pCALSR2XSB in frame with virus polypeptide to generate pCALSR2-VvPDS. The pCALSR2-AtFT plasmid possessing full-length coding sequence excluding the stop codon of the *AtFT* gene (AB027504, 525 bp) was reported previously [36]. The full-length coding sequence of the *VvFT* gene (525 bp, EF157728) was amplified with ‘VvFT-XhoI (+)’ and ‘VvFT-SmaI (−)’ primers (Appendix A) and introduced into the X/S/B site of pCALSR2XSB to generate pCALSR2-VvFT. The 201-base fragments of the *VvTFL1A* (nucleotide positions 71–271, DQ871591), *VvTFL1B* (nt positions 181–381, DQ871592), and *VvTFL1* (nt positions 223–413, DQ871593) genes were synthesized and used as a template to amplify fragments with primer pairs ‘VvTFL1A-XhoI (+)’ and ‘VvTFL1A- SmaI (−)’, ‘VvTFL1B- XhoI (+)’ and ‘VvTFL1B- SmaI (−)’, or ‘VvTFL1C-XhoI (+)’ and ‘VvTFL1C- SmaI (−)’ as described above (Appendix A). The fragments were each inserted into the X/S/B site of pCALSR2XSB to generate pCALSR2-VvTFL1A, B, C (Figure 1). A fragment of the *VvTFL1A* (201 bp, nt positions 71–271) and *VvTFL1B* (nt positions 429–611) genes were also amplified with primer pairs, ‘VvTFL1A-SalI’ and ‘VvTFL1A-MluI’, and VvTFL1B-SalI’ and ‘VvTFL1B-MluI’, respectively; excised at *Sal*I and *Mlu*I sites, and then introduced into the S-M site of pCALSR1SM to generate pCALSR1-VvTFL1A, B (Figure 1). 

### 2.3. Agro-Inoculation

Competent *A. tumefaciens* strain GV3101::pMP90 cells were transformed with RNA1 or RNA2 vector plasmids by electroporation with a Gene Pulser (Bio-Rad Laboratories, Hercules, CA, USA). Agrobacterium strain EHA101 expressing pBE2113::HC-Pro [38] was also used. After confirming the insertion sequences in ALSV vectors by colony PCR, agrobacterium strains were cultured in standard Lysogeny broth (LB) media overnight at 28 °C. Agrobacterium pellets were recovered after brief centrifugation and diffused in agro-inoculation buffer (10 mM MgCl_2_, 10 mM MES-KOH pH 5.7, 150 μM acetosyringone) at OD_600_ = 1.0. This suspension was kept at 22 °C in the dark for 2 h before inoculation. *N. benthamiana* (1–1.5 mo after sowing) was infiltrated with a mixture of three Agrobacterium strains expressing ALSV RNA1 and RNA2 clones, and an expression vector for the silencing suppressor gene *HC-Pro* as previously described [42]. Agro-inoculated *N. benthamiana* was kept in the dark overnight and grown for an additional two weeks. Virus infection was tested by RT-PCR analysis using upper non-inoculated leaves.

### 2.4. Preparation of RNAs from Infected Leaves

RNAs for inoculation by particle bombardment were prepared as follows. Upper leaves of infected *N. benthamiana* plants were harvested and stored at −80 °C, or immediately crushed with a mortar and pestle in liquid nitrogen. Infected leaves were homogenized in twice (*v/w*) the volume of extraction buffer (0.1 M Tris pH 7.8, 0.1 M NaCl, 5 mM MgCl_2_), and the extracts were used to inoculate *Chenopodium quinoa* leaves. Two to three weeks after inoculation, leaves with mosaic symptoms were collected and stored at −80 °C. In an ice-cold blender, infected *C. quinoa* leaves (10 g) were ground in 30 mL extraction buffer supplemented with 1% β-mercaptoethanol. Crude sap was roughly strained through two layers of cotton mesh and centrifuged at 10,233× *g* for 10 min at 4 °C to recover the supernatant. Extracts were clarified with bentonite and concentrated with PEG-6000, followed by RNA extraction with phenol-chloroform as reported previously [29,43]. The RNA was resuspended in water and the concentration was measured at 260 nm.

### 2.5. ALSV Inoculation by Particle Bombardment

After coating gold particles with RNA from infected leaves, grapevine seedlings (10–15 plants/each inoculation) were inoculated using the Helios Gene Gun system (Bio-Rad Laboratories, München, Germany) following a previously reported method [43]. The air pressure used was 1379 kilopascal (kPa) and each cotyledon was shot once with gold particles. Grapevine in vitro cultures were inoculated using the PDS-1000/He Particle Delivery System (Bio-Rad Laboratories, München, Germany) at 1379 kPa, with two shots per petri dish. The GDS-80 gene gun system (Nepa Gene Co., Ltd., Ichikawa, Japan) was also used for inoculation of in vitro cultures at 207 kPa. 

### 2.6. RT-PCR for Detection of Virus Infection

Total RNA was extracted from grapevine leaves three weeks after inoculation. Approximately 50 mg grapevine leaf tissue were frozen at −80 °C, crushed in a Micro Smash MS-100R (TOMY, Tokyo, Japan), and mixed with 600 µL extraction buffer (2% cetyltrimethylammonium bromide, 2% polyvinylpyrrolidone, 100 mM tris (hydroxymethyl) aminomethane, pH 8.0, 25 mM ethylenediaminetetraacetic acid, 2% β-mercaptoethanol). The homogenates were incubated at 65 °C for 20 min and mixed with 600 µL chloroform for 2 min. After centrifugation at 17,860× *g* for 10 min at 4 °C, the aqueous phase was recovered. One-third volume of 7.5 M LiCl was added to this solution, mixed well, and incubated at −80 °C for 30 min, −20 °C for 1 h, or 4 °C overnight. After centrifugation at 17,860× *g* for 30 min at 4 °C, RNA pellets were rinsed with 80% ethanol and dissolved in sterilized deionized water to a concentration of 1 µg µL^−1^. One microgram of RNA was reverse transcribed with ReverTra Ace (Toyobo, Osaka, Japan) following the manufacturer instructions, and PCR was performed with Ex Taq (Takara) using a Thermal Cycler Dice Version III (Takara, Kusatsu, Japan). The annealing temperature was 55 °C for the RT-PCR primers used (Appendix A). ALR2-999 (+) and ALR2-1437 (−) primers were used to detect wild-type ALSV (wtALSV). ALR2-1418 (+) and ALR2-1511 (−) were used to detect VvPDS, AtFT, and VvFT inserts. ALR1-6598 (+) and ALR1-6691 (−) were used to detect VvTFL1A insert.

### 2.7. Quantitative RT-PCR

Total RNA was extracted from grapevine cultures (three independent plants inoculated with ALSV-VvPDS) with the Plant/Fungi Total RNA Purification Kit (NORGEN, Thorold, ON, Canada). Extracted RNA was further treated with DNase I. A total of 500 ng RNA per 20 μL reaction was reverse transcribed with ReverTra Ace. Relative cDNA concentrations were quantified with the ECO Real-Time PCR System (Illumina, San Diego, CA, USA) using SYBR Premix Ex Taq II (Takara). Amplification of the target sequences was confirmed by melting curve analysis. The VvPDS gene was amplified with ‘VvPDS (+)’ and ‘VvPDS (−)’ primers (Appendix A), which targeted a 125-bp portion of VvPDS mRNA (JQ319631). A 128-bp portion of grapevine *Elongation factor 1α* gene (EF1α) was amplified with primers ‘VvEF1α (+)’ and ‘VvEF1α (−)’ (Appendix A) as an expression control. Expression of VvPDS was standardized with *EF1α*. The fluorescence signal of *EF1α* reached the threshold level within 20–24 cycles, indicating it is a good internal control.

### 2.8. Tissue-Blot Hybridization

Detection of ALSV infection in in vitro plant cultures was conducted with tissue-blot hybridization as described previously [33]. Briefly, grapevine plants were placed on positively charged nylon membranes (Hybond-N+; GE Healthcare, Tokyo, Japan), covered with plastic wrap, and frozen by pouring liquid nitrogen over the membranes. Plant tissues were crushed with a pastry pin to transfer plant exudates to the membranes. Membranes were soaked in 0.05 N NaOH for 30 min, washed in 20× SSC (saline-sodium citrate) buffer for 30 min, then RNA was fixed to the membranes by UV illumination. An RNA probe was hybridized to viral RNA and detected with an Image Quant LAS4000 (GE Healthcare) after reaction with CDP-Star (GE Healthcare) for 5 min. 

### 2.9. In Situ Hybridization

In situ hybridization analysis was conducted as described previously [37]. Shoots, tendrils, and flowers were excised from infected grapevine and fixed in formalin:ethanol:acetic acid:water (10:50:5:35 *v*/*v*), dehydrated with an ethanol/lemozol concentration series, and embedded in Paraplast Plus (Sigma-Aldrich Corp., St. Louis, MO, USA). Tissue sections of 12-μm thickness were prepared, extended on an APS-coated glass slide (Matsunami Glass, Osaka, Japan), deparaffinized, and hydrated. The sections were then treated with proteinase K and fixed once again. A DIG (digoxigenin)-labeled probe complementary to the Vp24 region of the ALSV-RNA2 [44] was used to hybridize slides. As a negative control, a DIG-labeled probe complementary to the P1 region of the soybean mosaic virus (SMV) genome was used [44]. The hybridized probes were labeled with sheep anti-DIG conjugated alkaline phosphatase (F. Hoffmann-La Roche SG, Basel, Switzerland) and stained with 5-bromo-4-chloro-3-indolyl phosphate (BICIP)/nitro blue tetrazolium (NBT) (F. Hoffmann-La Roche SG) to generate dark blue indigo dyes. The samples were dehydrated with an ethanol concentration series, dried, mounted in Entellan New (Merck KGaA, Darmstadt, Germany), and observed under a Leica MMLB optical microscope. 

### 2.10. Analysis of Sugar, Acid, and Anthocyanin Content in Grape Berries

Mature and colored berries were collected from early-flowered lines; Ganebu × ‘Nehelescol’ (240-1), ‘Cabernet Sauvignon’ × Ganebu (4-23), and Ganebu × ‘Shine Muscat’ (264-T31). Three individuals from each line were used for analysis. Berries of Ganebu, Yamabudo, and ‘Merlot’ were collected from field-grown grapevines. Commercial grape varieties used as control (‘Delaware’, ‘Kyoho’, and ‘Steuben’) were purchased from a fruit shop. The sugar and acid content of berries were measured with a sugar acidity meter (ATAGO Co., Ltd., Tokyo, Japan) following the manufacturer protocol. Total anthocyanin was analyzed as described by Shiozaki and Murakami [40]. Berry skin (2–5 mg) was incubated in 10 mL 50% (*v/v*) aqueous acetic acid at 4 °C overnight. Absorbance of the extracts at 530 nm was measured using a spectrophotometer (Novaspec Plus, Biochrom Ltd., Cambridge, UK). The amount of total anthocyanin was determined by comparing the sample absorbance to a standard curve from Malvidin 3-glucoside chloride (Funakoshi Co., Ltd., Tokyo, Japan).

### 2.11. High-Temperature Treatment for Virus Elimination

According to the procedure established in apple [36], infected F1 hybrid (G × N) seedlings grown to the 10 true-leaf stage at 25 °C were then incubated at 27 °C for 3 d, 30 °C for 5 d, 35 °C for 5 d, and then 37 °C for 30 d. After high-temperature treatment, the plants were grown at 25 °C until RT-PCR analysis. Five infected seedlings were subjected to the same high-temperature treatment and assayed for virus infection.

## 3. Results

### 3.1. Construction of ALSV Vectors for Grapevine

ALSV-RNA1 and -RNA2 sequences were introduced into binary expression vectors (accession ViralMultiSegProj15367) [36,42]. For cloning purposes, the RNA1 vector included a S/M site behind the polyprotein, while the RNA2 vector included a X/S/B site in the middle of the polyprotein (between MP and Vp25). These vectors were designated pCALSR1SM and pCALSR2XSB (Figure 1).

The infectious clones of ALSV vectors were first co-transformed into *N. benthamiana* by agro-inoculation [39] and infection was assayed with reverse transcription-polymerase chain reaction (RT-PCR). Virus was further propagated through rub-inoculation of *C. quinoa* (a propagation host) plants, and viral RNA was extracted from infected leaves. ALSV prepared from a combination of the empty vectors (pCALSR1SM and pCALSR2XSB) was designated as wild-type ALSV (wtALSV) for this report (Figure 1). To silence the *VvPDS* gene, a 201-bp fragment of *VvPDS* was introduced into the RNA2 vector (pCALSR2-VvPDS) and ALSV-VvPDS virus was prepared from pCALSR1SM and pCALSR2-VvPDS (Figure 1).

To express *AtFT* and grapevine *FT* (*VvFT*) in the ALSV vector, the full-length gene coding sequences were introduced into the RNA2 vector (pCALSR2-AtFT and pCALSR2-VvFT). The ALSV-AtFT and ALSV-VvFT vectors were prepared as a mixture of pCALSR1SM and pCLASR2-AtFT, and pCALSR1SM and pCLASR2-VvFT, respectively. To silence *VvTFL1A*, *VvTFL1B*, or *VvTFL1C*, 201-bp fragments of these genes were introduced separately into the RNA2 vector (pCALSR2-VvTFL1A, B, or C). ALSV-VvTFL1A, ALSV-VvTFL1B, or ALSV-VvTFL1C virus was prepared from pCALSR1SM and pCALSR2-VvTFL1A, B, or C (Figure 1). For simultaneous expression of *FT* and suppression of *VvTFL1A* or *B*, *A. tumefaciens* clones carrying pCALSR1-VvTFLA or B were mixed with those carrying pCALSR2-AtFT or pCALSR2-VvFT to generate ALSV-AtFT/VvTFL1A and ALSV-AtFT/VvTFL1B or ALSV-VvFT/VvTFL1A and ALSV-VvFT/VvTFL1B viruses (Figure 1).

All inserts in these viruses (e.g., VvPDS, VvTFL1A, B, and C, AtFT, and VvFT) were stably maintained after infection in *N. benthamiana*, *C. quinoa*, or grapevine plants, as determined by RT-PCR. The AtFT and VvFT inserts were sometimes deleted from virus RNA and the plants with deletions were eliminated from analysis.

### 3.2. ALSV Vector Inoculation Conditions for Grapevine Seedlings and In Vitro Cultures

*C. quinoa* and *N. benthamiana* are susceptible to ALSV and can be systemically infected at young or even mature growth stages [25]. However, for ALSV infection of fruit trees such as apple and pear, cotyledons of seedlings immediately after germination should be inoculated by bombardment with gold particles coated with concentrated viral RNA for efficient infection [36,43]. We first tested whether ALSV can infect grapevine seedling by particle bombardment. Progeny seedlings derived from *Vitis* spp. cv. ‘Koshu’ at three different growth stages were used in this analysis; seedlings immediately after germination with folded cotyledons, seedlings with expanded cotyledons, and seedlings with three true leaves (Figure 2A), were assessed in this experiment. The wtALSV or ALSV-VvPDS was used to inoculate cotyledons at the ‘folded cotyledon’ and ‘expanded cotyledon’ stages, or the first and second true leaves at the ‘three true leaves’ stage by particle bombardment. Local infection within inoculated leaves and systemic infection to upper leaves were assessed by RT-PCR. Table 1 and Figure 2B show that systemic infection was found in approximately 90% of seedlings when inoculated at the ‘folded cotyledons’ stage, whereas only 10% was infected when inoculated at the ‘expanded cotyledons’ stage, and no seedling was systemically infected when inoculated at the ‘three true leaves’ stage. Although systemic infection did not occur, all seedlings were locally infected in inoculated leaves at the ‘expanded cotyledons’ and the ‘three true leaves’ stages (Table 1), indicating that systemic movement of the virus was severely restricted in plants at these developmental stages.

Fruit trees usually have heterozygous genomic compositions and are propagated as clones through cutting or grafting. This means that genotypes and phenotypes of seedlings are usually different from the original variety from which they were derived. Therefore, ALSV infection of mature plants or in vitro cultured plantlets is desirable for genetic analysis and breeding of a specific variety. We prepared an in vitro culture of ‘Neo Muscat’, a table grape popular in Japan. After extension on media, vines of the ‘Neo Muscat’ culture were cut every two to three nodes to propagate. These cultures were inoculated by particle bombardment with wtALSV or ALSV-VvPDS at two different growth stages with a PDS-1000/He^TM^ system or a GDS-80 gene gun system. The first stage was within one to two weeks after cutting in which vines had not formed roots (Figure 2C, left). Vines were rooted at the second stage three to four weeks after cutting (Figure 2C, right). Whole plants, including vines and small expanding leaves, were inoculated in the first stage with a PDS-1000/He^TM^ system. In the second stage, the two uppermost leaves were inoculated with a GDS-80 gene gun system (Figure 2C, right). After assaying infection by tissue-blot hybridization (Figure 2D) or RT-PCR, we found that ALSV systemically infected 17–29% and 70% of in vitro cultures that were inoculated at the first stage (non-rooted) with a PDS-1000/He^TM^ system and the second stage (rooted) with a GDS-80 gene gun system, respectively. From these results, in vitro cultures at the true leaf stage had a high infection rate with a GDS-80 gene gun system (Table 1). 

Figure 3A shows a grapevine seedling from ‘Koshu’ infected with wtALSV three months post inoculation (mpi). The plant was asymptomatic but ALSV was detected in all (1st to 12th) true leaves by RT-PCR (Figure 3B). Grapevine seedlings or in vitro cultures infected with wtALSV did not show any discernible difference in appearance (i.e., shape and color) from non-infected plants. In situ hybridization analysis of shoot apical tissues showed that ALSV was distributed in leaf primordia of infected grapevine seedlings (Figure 3C). ALSV was also distributed throughout all tissues including the apical meristem cells of tendrils in infected plants (Figure 3D). 

### 3.3. VIGS in Grapevine Using the ALSV Vector

ALSV-VvPDS (Figure 1) was used to inoculate grapevine seedlings of ‘Koshu’ and Ganebu, and in vitro cultures of ‘Neo Muscat’ using particle bombardment as described above. Infection rates were similar to those inoculated with wtALSV (Table 1). All of the ‘Koshu’ seedlings infected with ALSV-VvPDS had a photo-bleaching phenotype caused by the loss of function of the *PDS* gene in portions of the first true leaf, and in the whole second or third true leaves as well as those that developed above these true leaves (Figure 4A), after which plant growth stopped. When in vitro cultures of ‘Neo Muscat’ were inoculated with ALSV-VvPDS, photo-bleaching first appeared along the veins of upper leaves in inoculated plants, spread to the upper leaves (Figure 4B), and then the leaves of the plants stopped growing. The average *VvPDS* mRNA accumulation decreased to 5% in white leaves, compared with green leaves and non-infected leaves, although there was variation in the calculated values (Figure 4C).

### 3.4. VIF in Grapevine Using the ALSV Vector

Natural flowering of grapevine seedlings generally requires a long period (several years) after germination. In this study, early flowering in grapevine was tested by infection of ALSV-AtFT, ALSV-VvFT, ALSV-VvTFL1A, B, or C, ALSV-AtFT/VvTFL1A or B and ALSV-VvFT/VvTFL1A or B viruses (Figure 1 and Table 2). Three out of eight ‘Koshu’ seedlings infected by ALSV-AtFT set flowers at the shoot apex 20–37 days post inoculation (dpi), with four to seven true leaves at flowering (Figure 5A). Floral buds also formed at the apices of axillary buds. In contrast, ALSV-VvFT and ALSV solely expressing *VvTFL1A*, *VvTFL1B*, or *VvTFL1C* inserts without *AtFT* (ALSV-VvTFL1A, ALSV-VvTFL1B, and ALSV-VvTFL1C) never induced precocious flowering (Table 2). Flowering rate was judged by the number of plants that set floral buds within three months of the experiments. Any seedlings or in vitro cultures infected with wtALSV alone did not set floral buds during the experiments (Table 2). Flowering was observed in seedlings infected with ALSV-AtFT, ALSV-AtFT/VvTFL1A, and ALSV-AtFT/VvTFL1B virus in 38–89% of ‘Koshu’ and 67–100% of Ganebu (Table 2). There seemed to be no difference in flowering rate and phenotypes among plants infected with ALSV-AtFT, ALSV-AtFT/VvTFL1A, or ALSV-AtFT/VvTFL1B. In vitro cultures of ‘Neo Muscat’ inoculated with ALSV-AtFT/VvTFL1A or ALSV-AtFT/VvTFL1B showed 69–71% flowering (Table 2).

### 3.5. Use of VIF for Evaluation of F_1_ Hybrids from Crossing between the Two Vitis Species

*V. ficifolia* var. ganebu (Ganebu) is expected to be a novel genetic source for breeding new varieties of grapevine because it has no endodormancy and a high anthocyanin content in the grape skins [39,40,41]. However, Ganebu is not suitable as a cultivated grapevine because it is a dioecious grapevine variety (Figure 5B). Early-flowering Ganebu seedlings in Table 2 separated into 17 male vs. 14 female plants out of 31 plants in total. In contrast, all seedlings from ‘Koshu’ and in vitro cultures of ‘Neo Muscut’ in Table 2 produced bisexual flowers.

At first, we applied VIF to the F_1_ hybrid progeny from crossing between the two *Vitis* species; Ganebu and ‘Nehelescol’ (G × N) for selection of seedlings with bisexual flowers. Flowering was observed in 28/36 (flowering/infected, 78%) seedlings infected with ALSV-AtFT/VvTFL1A in G × N seedlings. Among the flowered seedlings, 46% (13/28) of plants produced bisexual flowers (Figure 5C right). 

Subsequently, we inoculated ALSV vectors to F_1_ hybrid lines of G × N (line 240-1), ‘Cabernet Sauvignon’ × Ganebu (CS × G, line 4-23), and Ganebu × ‘Shine Muscat’ (G × SM, line 264-T3), which were germinated aseptically and grown in plant-boxes on solid medium. These F_1_ hybrids infected with ALSV-AtFT/VvTFL1A or ALSV-AtFT/VvTFL1B set first floral buds with bisexual flowers 20–30 dpi and formed fruits successively in growth chamber conditions. Infected plants continued precocious flowering for several months after inoculation, then the plants stopped flowering. The flowers of infected plants were self-pollinated, and berries were formed as shown in Figure 6A,B. The grape berries ripened 6–10 months post inoculation and the diameter of the grape berries from G × N, CS × G, and G × SM were 0.9–1.0 cm, 0.7–0.9 cm, and 0.95–1.15 cm, respectively (Figure 6C), which was similar to that of Ganebu grown in field conditions (0.7–1.0 cm). 

To evaluate the quality of fruits on early-flowering F_1_ hybrids, we measured the sugar and acid contents of berries. The sugar content of F_1_ hybrids, G × N (240-1) and G × SM (264-T3) were almost the same as those of Ganebu and commercial grapevine varieties (i.e., ‘Kyoho’ and ‘Steuben’) (Appendix A). In contrast, the acid content of G × N (240-1), CS × G (4-23), and G × SM (264-T3) was higher than that of eating varieties ‘Delaware’, ‘Kyoho’, and ‘Steuben’, and similar to that of Ganebu (Appendix A). Total anthocyanin levels in the skins of F_1_ hybrids, G × N (240-1) and CS × G (264-T3), were slightly lower than that of Ganebu, but higher than those in the skins of the varieties ‘Delaware’, ‘Merlot’, ‘Kyoho’, and ‘Steuben’ (Appendix A, Figure 6D). 

As described above, it was possible to select the seedlings with bisexual flowers in a short period of time. The VIF using ALSV vectors also made it possible to analyze the compounds of fruits on F_1_ grapevine seedlings within a year after germination. Most grapes had viable seeds, which germinated and grew as F_2_ seedlings.

### 3.6. No Seed Transmission to Progeny Seedlings and Virus Elimination from Infected Grapevine Plants

In infected apple plants, ALSV is distributed in pollen grains, ovaries, and ovules of flowers, and infected apple can transfer ALSV to their progeny at a seed transmission rate of 4.5% [37]. In contrast, ALSV was not present in pollen and ovules in gentian plants and not transmitted to the gentian progeny plants [41]. In the present study, we investigated ALSV distribution in the flower organs of grapevine (Ganebu and F_1_ hybrid of G × N) that was infected with ALSV-AtFT/VvTFL1B. The in situ hybridization analysis indicated that ALSV was present in the anther wall, pollen grain, and ovule of flowers of infected plants (Figure 7A,B). This result is consistent with those reported for infected apple plants [37]. As described above, mature fruits from F_1_ plants contained viable seeds that germinated and grew into F2 seedlings. Therefore, we tested whether ALSV was transmitted to the F_2_ progeny seedlings from early-flowering F_1_ grapevine plants infected with ALSV-AtFT/VvTFL1B using RT-PCR and qRT-PCR [41]. ALSV was not detected from a total of 60 progeny F2 seedlings tested (data not shown), suggesting that ALSV was not transmitted to progeny seedlings from infected grapevine plants. This indicates that the F2 seedlings could be used for subsequent breeding plan as ALSV-free stocks. 

The incubation of ALSV-infected apple seedlings at high-temperature (37 °C) for four weeks could disable virus movement to newly growing tissue to obtain ALSV-free shoots from infected trees [36]. We investigated whether heat treatment could eliminate ALSV from newly growing tissue in grapevines. Infected F_1_ hybrid (G × N) seedlings were grown until the 10 true-leaf stage at 25 °C (Figure 8A) and then incubated at 27 °C for 3 days, 30 °C for 5 days, 35 °C for 5 days, and 37 °C for 30 days (Figure 8B). The plants developed newly about 10 true leaves for incubation at 27–37 °C (Figure 8B). Subsequently, the plants were further grown at 25 °C for 30 days and the presence of ALSV was assessed in all expanded leaves. Clear PCR product bands were detected until the 4th to 10th leaves, which had developed before high-temperature treatment, and faint bands were detected in the 11th to 15th leaves (Figure 8B). In contrast, no ALSV bands were detected in the 16th to 24th leaves even after incubation at 25 °C for 30 days following the high-temperature treatment (Figure 8B). The same results were obtained in all five plants treated with high temperatures. Detection of RT-PCR ALSV products was repeated after two, four, and six months, but no virus was detected in any upper leaves of all plants. From these results, high-temperature incubation of ALSV-infected grapevine seedlings inhibited the systemic movement of ALSV from infected tissues to newly developed leaves, and once long-distance movement of the virus was inhibited, ALSV could not infect the upper leaves. 

## 4. Discussion

The VIF and VIGS technologies using the ALSV vector are powerful tools for inducing early flowering and reducing generation time, as well as functional gene analysis in fruit trees [24,26,45]. If the technology is applicable in grapevine plants, it would be very useful for efficient breeding of new grapevine varieties.

To use ALSV vectors, it is important to establish an efficient inoculation method in the target plant species because it is generally difficult to efficiently infect fruit trees with viruses. Our results suggest that there is a significant difference in ALSV infection rates between three developmental stages of juvenile grapevine seedlings (Table 1 and Figure 2A). Only the ‘folded cotyledons’ stage was sensitive to systemic ALSV infection (Table 1). Although viral infection was detected in inoculated leaves at the ‘expanded cotyledons’ and ‘three true leaves’ stages, there was no evidence of systemic infection. These data indicated that ALSV is swiftly transferred from cotyledons to true leaves at the ‘folded cotyledons’ stage but does not undergo long-distance movement once cotyledons have expanded. Similar phenomena were observed in apple and pear; ALSV can infect inoculated leaves with particle bombardment but does not move into upper uninoculated leaves [43]. This limitation of viral movement may be a common feature in woody plants; however, once systemic infection was established, ALSV distributed throughout the plants without any symptoms after three months (Figure 3A,B). In situ hybridization analysis indicated ALSV invaded the shoot apical meristem and leaf primordia (Figure 3C,D), which was similar to ALSV distribution reported in apple plants [24,35].

It was worth investigating whether grapevine plants cultured in vitro could be infected with ALSV similarly to seedlings because this system is beneficial for the genetic analysis of preexisting varieties and for breeding new varieties. We found that in vitro cultures of ‘Neo Muscat’ with expanded true leaves (Figure 2C) were systemically infected with ALSV (Table 1 and Figure 2D). General differences in physiological states between soil-cultivated plants and tissue-cultured plants may exist that renders in vitro plants more susceptible to viral infection (Figure 2A,C). Indeed, there are differences in the expression of genes associated with natural immunity and/or disease resistance between tissue- and soil-cultured plants, such as those observed between ALSV and anthracnose infection resistance in soil-cultivated and tissue-cultured strawberry plants [23,46].

VIGS of the *VvPDS* gene in grapevine was success in this study (Figure 4). The upper-most leaves of infected grapevines were photo-bleached and mRNA levels of *VvPDS* were reduced to 5% of green leaves (Figure 4C). This suppression level was comparable to a previous study performed in apple, where the apple rubisco small subunit gene was silenced using ALSV [36]. Thus, ALSV can drive gene silencing in grapevine as well as in its original host plant. Photo-bleaching was observed not only in leaf blades, but also in petioles and vines of many grapevine plants infected with ALSV-VvPDS. Asymptomatic infection with ALSV can also facilitate analysis of functionally unknown genes in grapevine. So far, two virus vectors from GVA and GLRaV-2 have been reported to induce gene silencing in grapevine [20,21]. Both vectors were used to silence *PDS* or magnesium chelatase genes, and successfully induced photobleaching in grapevine leaves.

We have already reported early flowering in apple and pear with the expression of *AtFT*, which is known as ‘florigen’ [25,26,36,45]. Simultaneous expression of *AtFT* and suppression of *MdTFL1* improves early-flowering rates in apple seedlings [26]. Flowering in grapevine was induced approximately one month after inoculation of germinating ‘Koshu’, Ganebu, and Ganebu x ‘Nehelescol’ seedlings (Table 2 and Figure 5). Although accumulation of the AtFT protein was not examined in this study, AtFT was most likely expressed with these vectors based on the amplification of the *AtFT* insert from viral RNA isolated from infected plants, and the early-flowering phenotype observed in infected grapevine. Additionally, no flowering was observed in uninoculated plants or plants infected with wtALSV. Introduction of *VvFT* or *VvTFL1 A, B*, or *C* [47] alone with ALSV did not induce flowering in ‘Koshu’ and Genebu seedlings (Table 2). Although the reason that expression of *VvFT* does not induce flowering in grapevine is unknown, the same phenomenon occurs with a homologous combination of *MdFT1* or *2* from apple [26]. There is probably a complex mechanism in which flowering in grapevine cannot be induced only by expression of *VvFT* or suppression of *VvTFLs* expression, for example, other genetic factors are needed to be supplemented. 

Flowering was induced one month after inoculation with ALSV-AtFT, ALSV-AtFT/VvTFL1A, and ALSV-AtFT/VvTFL1B of germinating seedlings and in vitro cultures. We expected that the concurrent expression of *AtFT* and suppression of grapevine *VvTFL1A* or *B* would improve flowering rates in grapevine seedlings, similar to apple infected with ALSV-AtFT/MdTFL1 [26]. However, there was no difference in flowering rates among plants expressing ALSV-AtFT, ALSV-AtFT/VvTFL1A, and ALSV-AtFT/VvTFL1B (Table 2), suggesting that early flowering is due to *AtFT* gene expression, not concurrent expression of *AtFT* and suppression of grapevine *VvTFL1A* or *B*. 

When germinating seedlings were used for inoculation experiments, the plants flowered after four to five true leaves developed (Figure 5A) and continued flowering but could not grow enough to bear fruits with seed. Only small fruits set on parts of early-flowering plants, without viable seeds (Figure 5A). This could be due to the activity of *AtFT*, which induces grapevine flowering but changes all buds to flowers and terminates vegetative growth. However, in vitro cultures inoculated with ALSV vectors continued to grow and flower (Figure 5C), and consequently grew large enough to bear fruits (Figure 6 A,B). 

Ganebu is a wild grape endemic to the subtropical southwestern islands of Japan [39,41]. It has no endodormancy and has high anthocyanin content in its grape skins and could be a novel source of flavonoids [39,40]. Therefore, a breeding program to develop new grapevine varieties with ever-bearing phenotypes could utilize crosses between Ganebu and other grapevine varieties. In this paper, the ALSV vector was successfully used to promote early flowering in seedlings of F_1_ hybrids from Ganebu × ‘Nehelescol’, ‘Cabernet Sauvignon’ × Ganebu, and Ganebu × ‘Shine Muscat’. This technique would be readily available for the rapid selection of seedlings with bisexual or unisexual flowers (Figure 5B,C). The early flowering F_1_ hybrid plants (G × N, CS × G, and G × SM) with bisexual flowers produced fruits through self-pollination and the grapes were used for quality analysis. Comparison of the sugar, acid, and total anthocyanin content in Ganebu, F_1_ hybrid berries (G × N, CS × G, and G × SM), a wild grapevine (*Vitis coignetiae*) and commercial varieties of grapevine showed that both G × N (240-1) and CS × G (4-23) F_1_ hybrids had high anthocyanin and acid content comparable to those measured in Ganebu. Conversely, G × SM (264-T31) had low anthocyanin content compared with those of G × N (240-1) and CS × G (4-23) (Appendix A). Thus, the VIF technology using ALSV vectors made it possible to analyze the quality of fruits on grapevine seedlings within a year after germination.

Mature fruits from F_1_ plants contained viable seeds that germinated and grew into seedling plants. About 60 self-pollinated F_2_ seedlings from G × N crosses were checked for ALSV seed transmission. Although ALSV invaded pollen and ovules of infected parent plants (Figure 7), all F_1_ seedlings were not infected with ALSV. These results suggest that grapevine plants can block or minimize ALSV seed transmission by inhibiting viral propagation in embryos [44]. Currently, the F_2_ seedlings from G × N crosses being cultivated to select plants with ever-bearing phenotypes.

ALSV elimination from infected grapevine plants by high-temperature treatment was investigated as previously reported in apple [36]. High-temperature treatment of infected seedlings at 37 °C for 30 days inhibited ALSV systemic movement into newly developed leaves, even after incubation at 25 °C for 30 days (Figure 8). Once the multiplication and movement of the virus was arrested, ALSV was not longer able to infect upper leaves, which were virus-free (Figure 8). Thus, ALSV-free shoots were easily obtained from infected grapevines and used as breeding-stocks.

## 5. Conclusions

The availability of ALSV vectors for VIF was demonstrated through this research and will be useful for accelerating the breeding of new varieties of grapevine, combined with marker-assisted selection [48]. Recently, the new plant breeding technique for gene modification using the CRISPR-Cas9 system was also applied in plants including grapevine [49,50,51,52]. Our ALSV system will provide a helpful tool to shorten the time necessary for evaluating fruit qualities in gene-edited grapevine and eliminate exogenous genes from transformed plants by genetic crosses. 

## Figures and Tables

**Figure 1 viruses-12-00070-f001:**
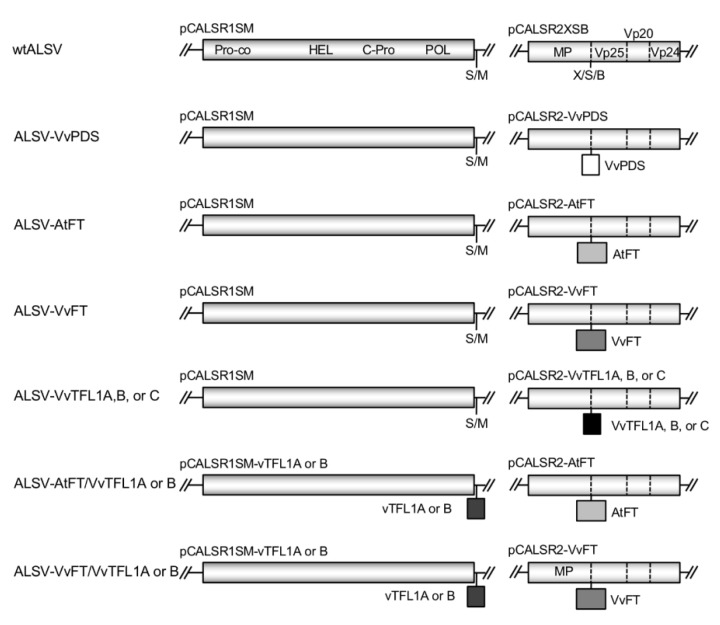
Schematic representation of infectious apple latent spherical virus (ALSV)-RNA clones (pCALSR1SM and pCALSR2XSB) and ALSV vectors containing a part of the phytoene desaturase (PDS) gene from grapevine (ALSV-VvPDS), full-length *FT* gene (*AtFT* from *A. thaliana* or *VvFT* from grapevine), and part of the *VvTFL A*, *B*, or *C* gene from grapevine. S/M and X/S/B indicate cloning sites of *Sal* I/*Mlu* I and *Xho* I/*Sam* I/*Bam* HI, respectively.

**Figure 2 viruses-12-00070-f002:**
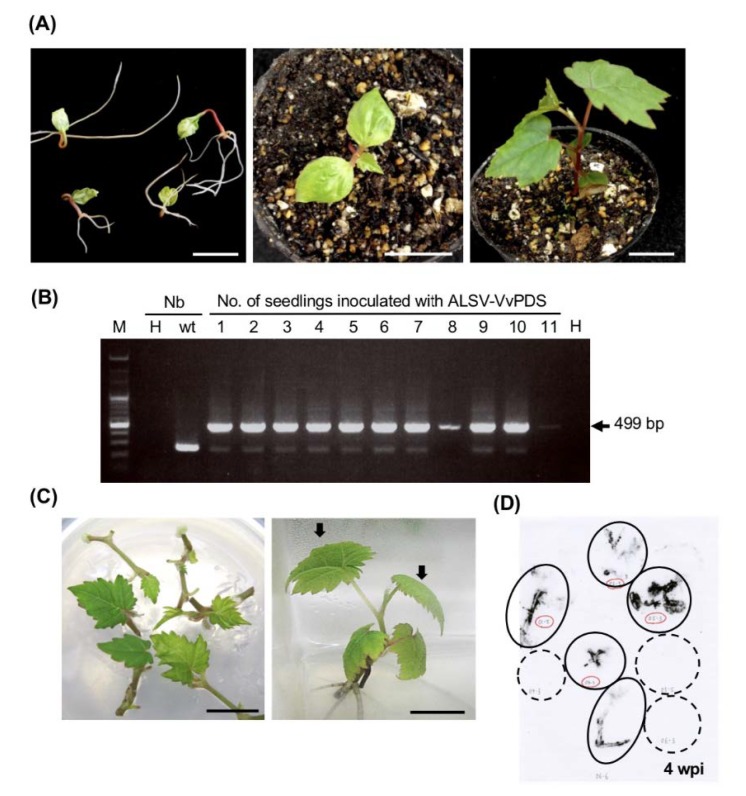
Inoculation of grapevine seedlings and in vitro cultures with ALSV. (**A**) Three growth stages of grapevine seedlings (‘Koshu’) used for ALSV vector inoculation. Left: ‘Folded cotyledons’, center: ’Expanded cotyledons’, and right: ‘Three true leaves’ stages. (**B**) RT-PCR detection of ALSV-VvPDS in grapevine seedlings inoculated at the ‘folded cotyledons’ stage. The third true leaves were assayed. Nb, *N. benthamiana*; H, non-inoculated healthy plant; wt; wtALSV-infected plant. (**C**) Three true leaf stages of in vitro cultures (‘Neo Muscat’) used for ALSV vector inoculation in non-rooted (**left**) and rooted (**right**) cultures. (**D**) Tissue blot hybridization of cultures inoculated with wtALSV in the non-rooted culture by PDS-1000/He™ system. Circles and dotted circles indicate infected and uninfected plantlets, respectively.

**Figure 3 viruses-12-00070-f003:**
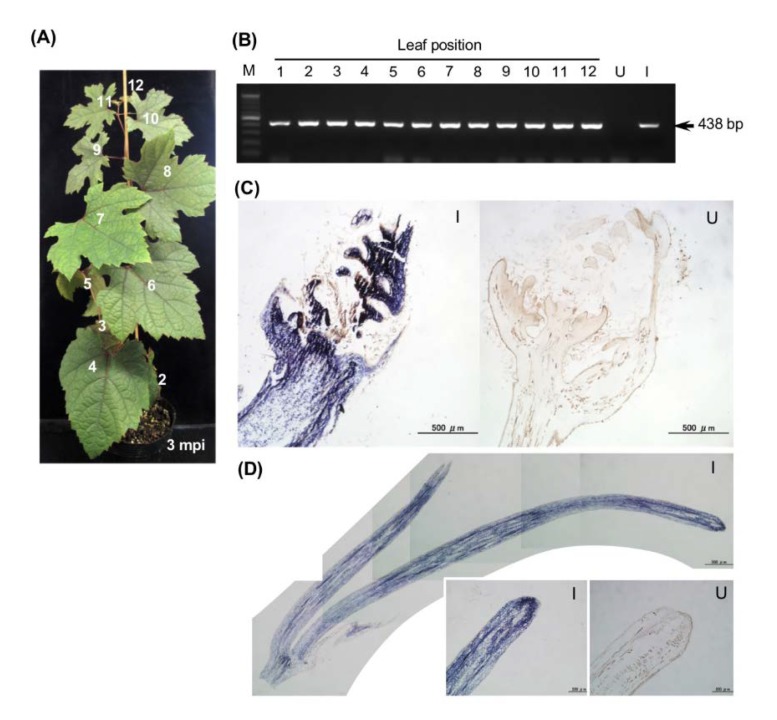
Systemic distribution of wtALSV in a grapevine seedling inoculated at the ‘folded cotyledons’ stage. (**A**) An infected grapevine seedling without visible symptoms three months post inoculation (mpi). Numbers indicate leaf positions used for detection of infection in (**B**). (**B**) Detection of wtALSV by RT-PCR from true leaves from the grapevine seedling shown in (**A**). M, DNA size maker; U, uninfected sample; I, infected sample. (**C**) In situ hybridization analysis of shoot tips from infected (I, left) and uninfected (U, right) grapevine seedlings using an ALSV-Vp24 (−) probe. (**D**) In situ hybridization analysis of tendrils of infected (I) and uninfected (U) grapevine seedlings probed with ALSV-Vp24 (−). Blue colour indicates ALSV distribution.

**Figure 4 viruses-12-00070-f004:**
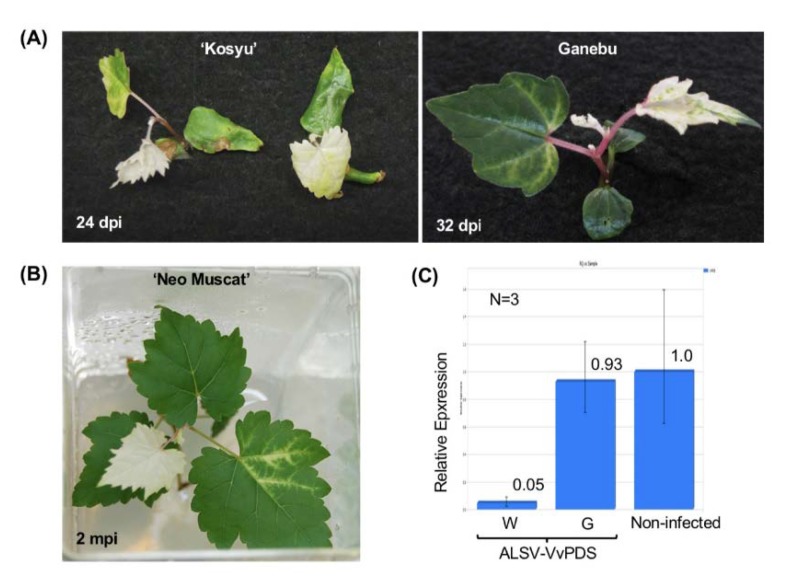
Photo-bleaching phenotype of grapevine seedlings and in vitro cultures inoculated with ALSV-VvPDS. (**A**) ‘Kousyu’ and Ganebu seedlings infected with ALSV-VvPDS at the ‘folded cotyledons’ stage. Dpi; days post inoculation. (**B**) A plantlet from tissue culture (‘Neo Muscat’) inoculated with ALSV-VvPDS at the true leaf stage as shown in Figure 2C (rooted culture) two months after inoculation (mpi). (**C**) A quantitative analysis of VvPDS-mRNA in white (W) and green (G) leaves of plants infected with ALSV-VvPDS as shown in (**B**). Reference gene: *VvEF1α.*

**Figure 5 viruses-12-00070-f005:**
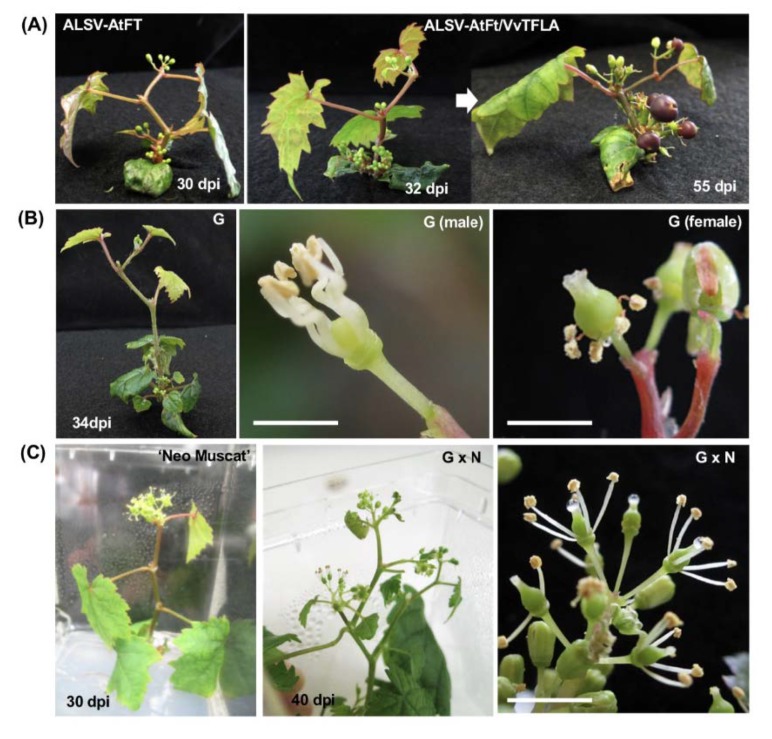
Early flowering of grapevine seedlings and in vitro cultures infected with ALSV-AtFT or ALSV-AtFT/VvTFL1A. (**A**) ‘Koshu’ seedlings inoculated at the ‘folded cotyledons’ stage. Dpi, days post inoculation. (**B**) A Ganebu seedling infected with ALSV-AtFT/VvTFL1A (left). Male (center) and female (right) flowers from Ganebu seedlings infected with ALSV-AtFT/VvTFL1A. Scale bars represent 2 mm. (**C**) Precocious flowering of in vitro cultures of ‘Neo Muscat’ (left) and Ganebu × ‘Nehelescol’ (G × N, line 240-1) (center) inoculated with ALSV-AtFT/VvTFL1A at the true leaf stage. Bisexual flowers from G × N are shown in the right-hand picture. A scale bar represents 5 mm.

**Figure 6 viruses-12-00070-f006:**
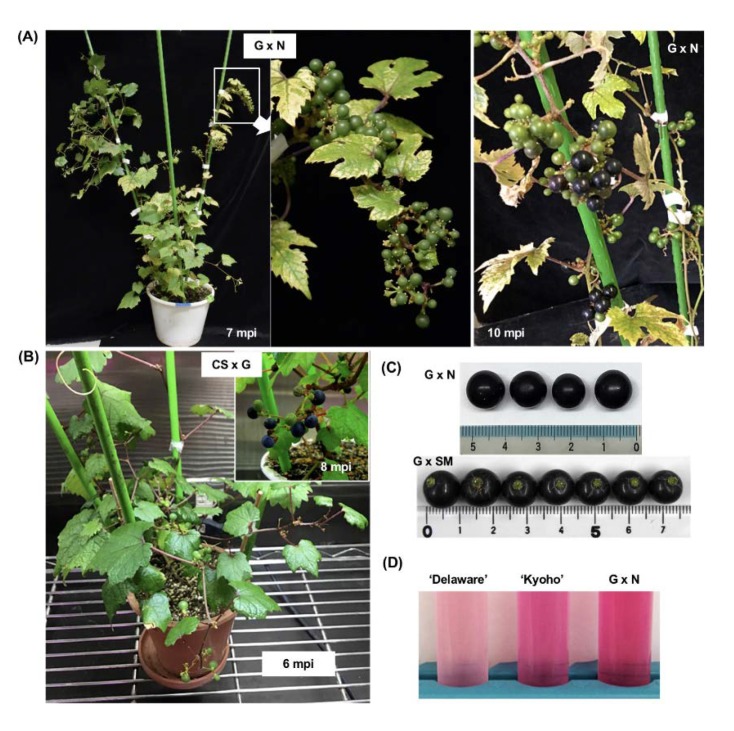
Fruiting in the F_1_ hybrid progeny from crossings between the *Vitis* species; Ganebu × ‘Nehelescol’ (G × N), ‘Cabernet Sauvignon’ × Ganebu (CS × G) and Ganebu × ‘Shine Muscat’ (G × SM) inoculated with the ALSV vector; and analysis of their grape berries. (**A**) Fruiting in a G × N (line 240–1) plant inoculated with ALSV-AtFT/VvTFL1A. (**B**) Fruiting in a CS × G (line 4–23) plant inoculated with ALSV-AtFT/VvTFL1B. (**C**) Grapevine berries produced by F_1_ hybrids of G × N and G × SM. (**D**) Comparison of total anthocyanin content in grape berries from G × N and the commercial varieties ‘Delaware’ and ‘Kyoho’.

**Figure 7 viruses-12-00070-f007:**
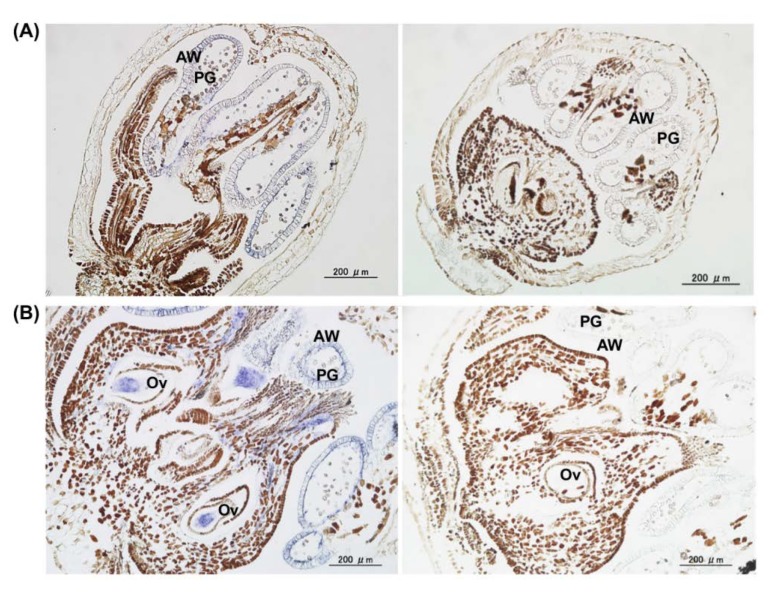
In situ hybridization analysis of ALSV distribution in the flower organs of Ganebu and F_1_ hybrid (G × N, line 240-1) plants infected with ALSV-AtFT/VvTFL1A. (**A**) Male flower of Ganebu treated with the ALSV-Vp24 (−) probe (left) and SMV-P1 (−) probe (right) as a control. (**B**) ALSV distribution in bisexual flowers in the F_1_ hybrid (G × N) detected with an ALSV-Vp24 (−) probe (left) and SMV-P1 (−) probe (right) as a control. Blue colour indicates ALSV distribution. AW; anther wall, PG; pollen grain, Ov; ovary.

**Figure 8 viruses-12-00070-f008:**
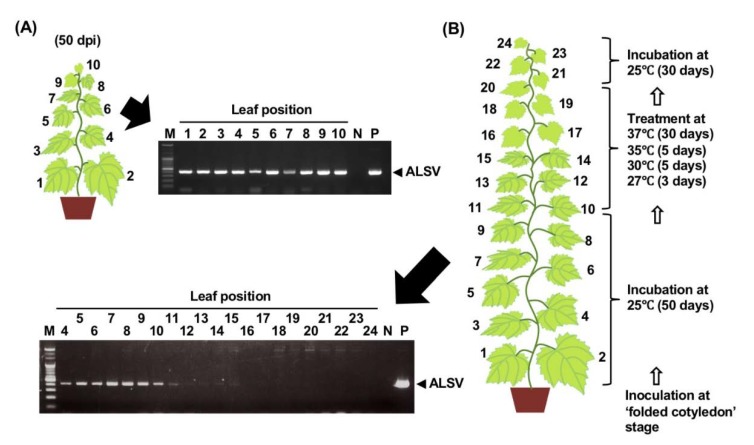
RT-PCR detection of ALSV in F_1_ hybrid G × N seedlings before and after high-temperature treatment. (**A**) ALSV infection in inoculated seedlings was assayed by RT-PCR at 50 dpi. (**B**) Infected plants were incubated at 27 °C for 3 days, 30 °C for 5 days, 35 °C for 5 days, and then 37 °C for 30 days. The seedlings were then grown at 25 °C for 30 days and all expanded leaves were analyzed by RT-PCR to assess ALSV infection. Numbers indicate the leaf position on a grapevine seedling. M, size maker; N, negative control (non-infected tissue); P, positive control (infected tissue).

**Table 1 viruses-12-00070-t001:** Infection of wild-type ALSV (wtALSV) and ALSV-VvPDS to the seedlings and in vitro cultures of grapevine by particle bombardment.

Material (Cultivar)	Growth Stage	System for Particle Bombardment *	ALSV Vector	No. of Infected/Inoculated Plants (%)	
				Local **	Systemic ***
Seedlings (‘Koshu’)	Folded cotyledons	Helios Gene Gun	wtALSV	nt	11/12 (92)
	Expanded cotyledons	Helios Gene Gun	wtALSV	10/10 (100)	1/10 (10)
	Three true leaves	Helios Gene Gun	wtALSV	10/10 (100)	0/15 (0)
	Folded cotyledons	Helios Gene Gun	ALSV-VvPDS	nt	10/11 (91)
	Expanded cotyledons	Helios Gene Gun	ALSV-VvPDS	10/10 (100)	1/12 (8)
	Three true leaves	Helios Gene Gun	ALSV-VvPDS	15/15 (100)	0/15 (0)
Plants cultured in vitro (‘Neo Muscat’)	True leaf, non-rooted	PDS-1000/He™	wtALSV	nt	17/58 (29)
	True leaf, rooted	GDS-80	wtALSV	nt	7/10 (70)
	True leaf, non-rooted	PDS-1000/He™	ALSV-VvPDS	nt	17/98 (17)

* Helios Gene Gun and PDS-1000/He™ systems are by Bio-Rad Laboratories, Inc. and GDS-80 is a gene gun system by Nepa Gene Co., Ltd. ** Infection was found on inoculated leaves. nt; not tested. *** Infection was found on upper leaves.

**Table 2 viruses-12-00070-t002:** Precocious flowering of the seedlings and in vitro cultures of grapevine infected with ALSV vectors possessing *FT* and/or *VvTFL1* sequences *.

Materials (Cultivar)	ALSV Vectors	No. of Flowered/Infected Plants (%)
Seedlings (‘Koshu’)	wtALSV	0/10 (0)
	ALSV-AtFT	3/8 (38)
	ALSV-VvFT	0/6 (0)
	ALSV-VvTFL1A	0/9 (0)
	ALSV-VvTFL1B	0/6 (0)
	ALSV-VvTFL1C	0/5 (0)
	ALSV-AtFT/VvTFL1A	8/9 (89)
	ALSV-AtFT/VvTFL1B	5/9 (56)
	ALSV-VvFT/VvTFL1A	0/6 (0)
	ALSV-VvFT/VvTFL1B	0/3 (0)
Seedlings (Ganebu)	wtALSV	0/10 (0)
	ALSV-AtFT	6/6 (100)
	ALSV-VvFT	0/6 (0)
	ALSV-VvTFL1A	0/10 (0)
	ALSV-VvTFL1B	0/10 (0)
	ALSV-VvTFL1C	0/5 (0)
	ALSV-AtFT/VvTFL1A	8/12 (67)
	ALSV-AtFT/VvTFL1B	17/21 (81)
	ALSV-VvFT/VvTFL1A	0/5 (0)
	ALSV-VvFT/VvTFL1B	0/5 (0)
*In vitro* cultures (‘Neo Muscut’)	wtALSV	0/20 (0)
	ALSV-AtFT/VvTFL1A	11/16 (69)
	ALSV-AtFT/VvTFL1B	5/7 (71)

* Seedlings were inoculated at folded cotyledon stage by a Helios gene gun system (Bio-Rad). The GDS-80 gene gun system was used for inoculation of in vitro cultures.

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
