# Peer review of "Virus-Induced Flowering by Apple Latent Spherical Virus Vector: Effective Use to Accelerate Breeding of Grapevine"

_viruses, 2020, doi:10.3390/v12010070_

Round 1

Reviewer 1 Report

The manuscript by Maeda et al. investigates the use of ALSV vectors for Virus-induced flowering (VIF) in seedlings and in vitro cultures of grapevine. Wild type and recombinant ALSV viruses were able to induce a systemic infection in grapevine plants, promote flowering and produce grapes with viable seeds within a year. This VIF technology was successfully used on F1 hybrids and allowed to analyse phenotypic features of berries within a year after germination. Moreover, ASLV could be removed from grapevine tissue by heat treatment. Thus, VIF can be used to accelerate breeding procedures of grapevines with desired traits. In addition, the authors demonstrate that ALSV vectors are able to induce VIGS in grapevine. This results associated to a reliable inoculation procedure open the way to a user-friendly tools to gene functional analysis in grapevine.

The results presented on this manuscript are of importance. The presented data are convincing and the methods were clearly described. The manuscript is also well written. The whole study is sound and deserves publication.

Please find below some specific comments to improve the manuscript.

Introduction section

Line 64 : I suggest to replace “penetrates” by “invades”

Line 68-70: Could the authors precise the delay for blossom processes in a wild plant to get an idea of the gain of time?

Line 131 : could the authors precise the procedure for agrobacterium inoculation : infiltration , needle inoculation,….?

Line 133 : replace “Inoculated” by “Agro-inoculated”

Line 136 : Paragraph 2.4 : Might the authors modify the title of this paragraph to indicate that RNA extraction is dedicated to gold particle coating or add one sentence at the beginning of the paragraph to explain the objective of this RNA extraction?

Results

I suggest making two distinct sections to present VIGS and VIF results

VIF results: To facilitate the reading of the results, I would appreciate to get a table that summarize the flowering genes used in this study, their plant hosts, their expected functions, and the fragment used in this study. This table could be presented as supplemental data.

Line 345 : “Natural flowering of grapevine….” Please, give a reference for this statement.

Line 353 : “wt ALSV alone did not set floral buds..” This result is not presented in the data of the manuscript. I would suggest to provide this data in the appropriate tables.

Author Response

Dear Reviewer 1,

Thank you very much for reviewing our manuscript. Our incorporation of the Reviewer 1’s suggestions is as follows: Corrected words and sentences rea written in red.

Introduction section 

Line 64 : I suggest to replace “penetrates” by “invades”

> The word “penetrates” has been replaced to “invades”, as suggested.

Line 68-70: Could the authors precise the delay for blossom processes in a wild plant to get an idea of the gain of time?

> The sentences “Apple seedlings generally take at least 6-7 years from germination to flowering in the wild. However, using VIF,” have been added, as suggested.

Line 131 : could the authors precise the procedure for agrobacterium inoculation : infiltration , needle inoculation,….?

> The word “inoculated” has been changed to “infiltrated” as suggested.

Line 133 : replace “Inoculated” by “Agro-inoculated”

> The word “Inoculated” has been replaced to “Agro-inoculated”, as suggested.

Line 136 : Paragraph 2.4 : Might the authors modify the title of this paragraph to indicate that RNA extraction is dedicated to gold particle coating or add one sentence at the beginning of the paragraph to explain the objective of this RNA extraction?

> A sentence “RNAs for inoculation by particle bombardment were prepared as follows.” Has been added, as suggested.

Results 

I suggest making two distinct sections to present VIGS and VIF results

> VIGS and VIF have been separated to two distinct sections, as suggested.

VIF results: To facilitate the reading of the results, I would appreciate to get a table that summarize the flowering genes used in this study, their plant hosts, their expected functions, and the fragment used in this study. This table could be presented as supplemental data.

> We used only FT genes (Arabidopsis and Vitis) and TFL genes from Vitis for flowering experiments. The accession no. and sizes of the genes used are summarized in Table S1.

Line 345 : “Natural flowering of grapevine….” Please, give a reference for this statement.

> This is commonly referred years (periods), and there is no literature. The words “a long period (3-5 years)” have been changed to “a long period (several years)”

Line 353 : “wt ALSV alone did not set floral buds..” This result is not presented in the data of the manuscript. I would suggest to provide this data in the appropriate tables.

> The data on wtALSV alone have been added in Table 2, as suggested.

Best regards,

N. Yoshikawa

Reviewer 2 Report

The manuscript by Maeda et al., describes the application of virus-induced flowering technologies to grapevine. The group has reported similar systems in apple, pear and strawberry. As the authors describe, the technique has the potential to accelerate the breeding in woody perennial crops. The conclusions are well supported by the results, but the presentation needs some work. Also, the rationale for some of the experiments need to be better defined e.g., the experiments with VvFT and VvTFL1 (see below).

In addition to the comments below, a marked copy of the manuscript with additional corrections and comments has been uploaded.

In the abstract and on line 148 of the Materials and Methods, the authors say the plants were inoculated with RNA. However, the Results section (line 236) says that the clones were driven by the CaMV 35S promoter. Hence, it sounds like plants were inoculated with cDNA not with in vitro transcripts. The plasmids would be expected to direct synthesis of ALSV RNAs in vivo, but the plants would have been inoculated with cDNA.

Line 211: which chromogenic substrate was used?

This brings up another point. Some of the methods are described in the Results section. In addition to not mentioning the CaMV 35S promoter in the method section, the authors state on line 229 of the methods section that plants were grown “as described in the Results section”. All methods should be described in the methods section and do not need to be repeated in the Results section.

Abbreviations that are defined but not used do not need to be defined.

Full genus and species names should be given at first use.

Gene names should only be italicized for formally named gene. When phytoene desaturate and green fluorescent protein are used to refer to groups of related genes, they should not be italicized. Named genes should be capitalized and italicized, e.g., the Arabidopsis FLOWERING LOCUS T gene. According to Jung et al. (2015; Tree Genetics & Genomes 11:108), the species prefix for apple should be “Mdo” not “Md”.

All units should be metric – psi should be converted to kPa.

Suppliers only need to be listed once. If they are given in the methods section they do not need to be repeated in the Results section.

Section heading number “2.9” is repeated in the methods section

It is not clear why the authors thought that silencing VvTFL1A would induce flowering. According to Carmona et al. (2007), overexpression of VvTFL1A in Arabidopsis did not affect flowering time. Similarly, Carmona et al. (2007) reported that overexpression of VvFT induced early flowering in Arabidopsis. On line 507, the authors say that the reason why VvFT did not induce flowering in grapevine is unknown. However, wouldn’t insertion of VvFT into ALSV be expected to lead to VIGS of VvFT and not early flowering?

Author Response

Dear Reviewer 2,

Thank you very much for reviewing our manuscript. I really appreciate your comments and corrections on our manuscript.

Our incorporation of the Reviewer 1’s suggestions is as follows:

Corrected words and sentences rea written in red.

In addition to the comments below, a marked copy of the manuscript with additional corrections and comments has been uploaded.

> I have corrected everything according to your suggestions.

In the abstract and on line 148 of the Materials and Methods, the authors say the plants were inoculated with RNA. However, the Results section (line 236) says that the clones were driven by the CaMV 35S promoter. Hence, it sounds like plants were inoculated with cDNA not with in vitro transcripts. The plasmids would be expected to direct synthesis of ALSV RNAs in vivo, but the plants would have been inoculated with cDNA.

> For ALSV inoculation to fruit trees including grapevine, bombardment with gold particles coated with viral RNAs has been used for efficient infection. DNA clones could not be used for inoculation to grapevine directly, because of low infectivity. For this, we first inoculate the DNA clones driven by CaMV 35S promoter to N. benthamiana by agro-infiltration, and then the extracts of infected N. benthamiana leaves were inoculated to C. quinoa plants (a propagation host of ALSV). Then RNAs for inoculation were prepared from infected C. quinoa plants. Biolistic inoculation of RNAs from infected C. quinoa leaves is the only reliable inoculation method. 

Line 211: which chromogenic substrate was used?

> A sentence “5-bromo-4-chloro-3-indolyl phosphate (BICIP)/nitro blue tetrazolium (NBT) (F. Hoffmann-La Roche SG) to generate dark blue indigo dyes.” has been added, as suggested.

This brings up another point. Some of the methods are described in the Results section. In addition to not mentioning the CaMV 35S promoter in the method section, the authors state on line 229 of the methods section that plants were grown “as described in the Results section”. All methods should be described in the methods section and do not need to be repeated in the Results section.

> A sentence “In these vectors, RNA1 and RNA2 sequences were driven by cauliflower mosaic virus 35S RNA promoter and ended with the nopaline synthase terminator” was moved to 2.2 Constraction of ALSV vector in Materials and Methods.

Abbreviations that are defined but not used do not need to be defined.

> Abbreviations has been deleted as suggested.

Full genus and species names should be given at first use.

> Full genus and species name has been changed as suggested.

Gene names should only be italicized for formally named gene. When phytoene desaturate and green fluorescent protein are used to refer to groups of related genes, they should not be italicized. Named genes should be capitalized and italicized, e.g., the Arabidopsis FLOWERING LOCUS T gene. According to Jung et al. (2015; Tree Genetics & Genomes 11:108), the species prefix for apple should be “Mdo” not “Md”.

All units should be metric – psi should be converted to kPa.

> Units has been converted to kPa, as suggested.

Suppliers only need to be listed once. If they are given in the methods section they do not need to be repeated in the Results section.

    > Suppliers have been deleted in the Results section as suggested.

Section heading number “2.9” is repeated in the methods section

    > Section heading number has been corrected.

It is not clear why the authors thought that silencing VvTFL1A would induce flowering. According to Carmona et al. (2007), overexpression of VvTFL1A in Arabidopsis did not affect flowering time. Similarly, Carmona et al. (2007) reported that overexpression of VvFT induced early flowering in Arabidopsis. On line 507, the authors say that the reason why VvFT did not induce flowering in grapevine is unknown. However, wouldn’t insertion of VvFT into ALSV be expected to lead to VIGS of VvFT and not early flowering?

> From the results of other plant species, I do not think that insertion of VvFT into ALSV lead to VIGS of VvFT and not early flowering. We believe that a more complex mechanism exists for flowering in trees. So, we have added the sentence “Probably, there is a complex mechanism in which flowering in grapevine cannot be induced only by expression of VvFT or suppression of VvTFLs expression, for example, other genetic factors are needed to be supplemented”

Best regards,

N. Yoshikawa